# Plasma Metabolomics Identifies Lipid and Amino Acid Markers of Weight Loss in Patients with Upper Gastrointestinal Cancer

**DOI:** 10.3390/cancers11101594

**Published:** 2019-10-19

**Authors:** Janice Miller, Ahmed Alshehri, Michael I. Ramage, Nathan A. Stephens, Alexander B. Mullen, Marie Boyd, James A. Ross, Stephen J. Wigmore, David G. Watson, Richard J.E. Skipworth

**Affiliations:** 1Clinical Surgery, University of Edinburgh, Royal Infirmary of Edinburgh, Edinburgh EH16 4SA, UK; j.miller@ed.ac.uk (J.M.); mramage@exseed.ed.ac.uk (M.I.R.); j.a.ross@ed.ac.uk (J.A.R.); s.wigmore@ed.ac.uk (S.J.W.); 2The Royal Liverpool University Hospital, Prescot Street, Liverpool L7 8XP, UK; Ahmed.Alshehri@strath.ac.uk (A.A.);; 3Strathclyde Institute of Pharmacy and Biomedical Sciences, University of Strathclyde, Glasgow G1 1XQ, UK; a.mullen@strath.ac.uk (A.B.M.); marie.boyd@strath.ac.uk (M.B.); d.g.watson@strath.ac.uk (D.G.W.)

**Keywords:** cachexia, cancer, metabolomics, high resolution mass spectrometry

## Abstract

Cachexia is a multifactorial wasting syndrome associated with high morbidity and mortality in patients with cancer. Diagnosis can be difficult and, in the clinical situation, usually relies upon reported weight loss. The ‘omics’ technologies allow us the opportunity to study the end points of many biological processes. Among these, blood-based metabolomics is a promising method to investigate the pathophysiology of human cancer cachexia and identify candidate biomarkers. In this study, we performed liquid chromatography mass spectrometry (LC/MS)-based metabolomics to investigate the metabolic profile of cancer-associated weight loss. Non-selected patients undergoing surgery with curative intent for upper gastrointestinal cancer were recruited. Fasting plasma samples were taken at induction of anaesthesia. LC/MS analysis showed that 6 metabolites were highly discriminative of weight loss. Specifically, a combination profile of LysoPC 18.2, L-Proline, Hexadecanoic acid, Octadecanoic acid, Phenylalanine and LysoPC 16:1 showed close correlation for eight weight-losing samples (≥5% weight loss) and nine weight-stable samples (<5%weight loss) between predicted and actual weight change (r = 0.976, *p* = 0.0014). Overall, 40 metabolites were associated with ≥5% weight loss. This study provides biological validation of the consensus definition of cancer cachexia (Fearon et al.) and provides feasible candidate markers for further investigation in early diagnosis and the assessment of therapeutic intervention.

## 1. Introduction

Cancer cachexia has been defined as “a multifactorial syndrome characterized by an ongoing loss of skeletal muscle mass that cannot be fully reversed by conventional nutritional support and leads to progressive functional impairment” [1]. The agreed consensus diagnostic criterion for cachexia is either weight loss >5% over 6 months or weight loss >2% in individuals already showing signs of depletion (BMI < 20 kg/m^2^ or skeletal muscle index consistent with sarcopenia) [1]. Cancer cachexia is characterized by loss of adipose tissue, skeletal muscle, and appetite, and impacts negatively the quality of life of patients with cancer, response to treatment and survival [2]. Therefore, managing cachexia should be considered a central component of cancer patient treatment.

As cachexia develops progressively through various stages, from pre-cachexia to cachexia to refractory cachexia, the identification of biomarkers that relate to stage and diagnosis would be particularly important to prevent or delay deterioration [1]. Potential markers of cachexia may also have value through future studies as outcome measures of therapeutic intervention. Imaging methods such as CT and MRI are currently considered precise measures of body composition but have several limitations, including cost, availability, and exposure to radiation (in the case of CT) [3].

Recent progress in high-throughput analytical technologies and bioinformatics now permits simultaneous analysis of hundreds of compounds constituting the metabolome [4]. Metabolomic analyses give complex fingerprints that appear to be characteristic of a given metabolic phenotype or diet. Several previous studies have attempted to quantify urinary and plasma metabolites associated with cachexia. They have identified metabolites that are possibly discriminative of cachexia, indicating that there may be scope for clinical application of metabolomic biomarkers of cachexia [5,6,7,8,9].

We have previously analysed urinary proteomics as a measure of degradation products in the circulating fraction, allowing us to discriminate between weight-losing and weight stable cancer patients [10]. Building upon the theory that metabolites produced from tissue breakdown are likely to be found in plasma, we investigated whether we could use liquid chromatography mass spectrometry (LC/MS)-metabolomics to detect plasma metabolites associated with weight loss in upper GI cancer patients. In particular, we aimed to identify a metabolic signature that relates directly to patient weight loss. Plasma was selected as the biofluid of choice as it has been shown previously that several end products of muscle catabolism (e.g., creatinine and methylhistidine) can be easily measured within [11].

## 2. Results

Plasma samples were analysed from upper GI cancer patients (*n* = 18) taken from a cross-sectional cohort of upper GI cancer patients who were recruited to two previously-published studies of muscle transcriptomics (*n* = 65 pre-surgical rectus abdominis biopsies, and *n* = 12 pre- and post-surgical resection muscle biopsies) [12,13]. In these previous studies, quantitative significance analysis of microarrays produced an 83-gene signature that was able to identify patients with >5% weight loss, while this molecular profile was unrelated to markers of systemic inflammation. Comparison with healthy control muscle revealed that despite differences in the muscle transcriptome at baseline (941 genes regulated), the muscle of patients at post-surgical resection follow-up was similar to control muscle (two genes regulated). Baseline, pre-surgical plasma samples were available for 18 of these patients. Therefore, for the present study, there were nine patients who had experienced ≥5% weight loss in the previous 6 months (≥5% WL group), and nine patients with <5% weight loss (weight stable group: WS). Cancer cachexia had been primarily defined as ≥5% weight loss in order to identify patients who would have experienced dynamic wasting and thus be more likely to have identifiable markers and metabolic signatures. If the patients were analysed as two separate groups according to these weight loss criteria, the mean percentage weight loss in the ≥5% WL group was 14.39% compared with 2.13% in the WS group (*p* = 0.001). There were more males in the WS group. There was no significant difference in age, body mass index (BMI), Skeletal Muscle Index (SMI), Subcutaneous Adipose Tissue Index (SATI) or Visceral Adipose Tissue Index (VATI) between the groups. However, patients in the ≥5% WL group did demonstrate higher CRP levels compared with WS patients. These body composition and systemic inflammatory phenomena are all associated with worsened outcomes in cancer patients [14,15,16], confirming the ≥5% WL patients as a high-risk group. Two patients in the WS group had >2% weight loss and low SMI [according to Martin criteria [17] on the CT scan. Patients had a mixture of upper gastrointestinal cancers—predominantly those of the oesophagus and pancreas (Table 1). On the overall metabolomic analysis, a total of 40 metabolites were significantly associated with ≥5% weight loss according to univariate analysis using a T test. The metabolites with the highest fold change were L-phenylalanine and various species of LysoPE, LysoPA and LysoPC. These metabolites, as well as free fatty acid levels (FFAs), were all elevated in the ≥5% WL group.

Appendix A shows the principal components analysis (PCA-X) of all 18 samples. Three quality control (QC) samples were also included in the run and indicated that the instrument stability was good for the duration of the run. PCA-X, an unsupervised model in SIMCA-P, produces a natural scatter of the samples based on their characteristic metabolomics footprints. In general, there was no separation of samples according to the cachexia threshold defined as ≥ 5% weight loss (Appendix A).

Supervised models enable identification of metabolites that have the most significant contribution to a given clustering pattern. In SIMCA, supervised analysis can be carried out using orthogonal partial least squares discriminant analysis (OPLS-DA) or orthogonal partial least squares (OPLS) models. An OPLSDA model was not very successful in classifying the samples without the need to omit several. However, an OPLS model based on six metabolites (Table 2) was successfully produced for all but one of the patients who had ≥5% weight loss. This individual patient sample was omitted from the OPLS model since it clustered with the WS samples. The model plotted predicted against actual weight loss, and had a CVANOVA of 0.0014. The correlation line had an r value of 0.976 when fitted through the samples (Figure 1). Therefore, these six metabolites provide a useful metabolomic signature for further longitudinal testing.

Table 3 shows the metabolites that were found to be significantly different between the two weight loss categories. The ratio represents the intensity of the metabolites relative to the WS patient group.

Figure 2 is a heat map showing the relative abundance of the lyso- lipids in these plasma samples. Lyso-PC18:2 was almost as abundant as Lyso-PC 16:0 and was elevated by 1.75 fold. Beyond these two lyso-lipids, the metabolite abundance was much lower but there were many more minor lipids showing similar or greater fold changes in the ≥5% WL group compared with WS.

## 3. Discussion

In this study, we performed LC-MS based metabolomics analysis to reveal the metabolic profile of weight loss in cancer. We were able to demonstrate distinct profiles associated with the presence or absence of ≥5% WL. Most of the metabolites identified within these profiles fell within the lipid pathways. The clearest finding was that several long chain fatty acids and lysolipids were elevated in the plasma of the patients with higher weight loss. Lysolipids are very abundant in plasma [18] and Lyso-PC 16:0 was the most abundant compound by response in this set of samples; an increase of 1.34 fold between ≥5% WL and WS patients represents a major shift in metabolic output.

The OPLS model shown in Figure 1 indicates a strong association between weight loss and six metabolites. Two of the metabolite markers were lysolipids, two were amino acids, and two were fatty acids. There is a risk of overfitting data when the numbers of samples are small. However, in the present study, there was a clear indication that a small number of markers may be used to model the degree of weight loss with accuracy. One patient with ≥5% WL was removed from the analysis as this sample clustered with the WS group, but for this small cohort alone, our six identified metabolites, as a diagnostic test for cachexia (≥5% WL), would be 95% accurate. Validation of these markers will require larger studies, ideally with sequential assessment. Therefore, the current study demonstrates an association between lipolytic activity in the plasma of cancer patients with weight loss. The changes in the amino acids found add credence to the importance of muscle wasting in cancer cachexia and indeed, most current research into cancer cachexia focuses on this area. However, the importance of lipid metabolism is re-emerging as an area of priority [19].

Previously, cachexia in patients with cancer of the oesophagus and pancreas has been linked with high levels of plasma glycerol and free fatty acids [20,21]. Weight-losing cancer patients have been shown to have increased turnover of both glycerol and fatty acids compared with cancer patients without weight loss [22]. Some, however, have suggested that observed increases in lipolysis and triglyceride-fatty acid cycling in cachectic patients with oesophageal cancer are due to alterations in nutritional status rather than disease presence [23]. Cachectic ovarian cancer patients have been shown to have increased levels of free fatty acids, monoacylglycerides and diglycerides in their serum and ascitic fluid [24]. Whilst it is difficult to determine where fatty acid and lipid metabolites originate, both lysolipids and fatty acids are markers of lipolysis [25,26]. Free fatty acids may also provide energy for the tumour, with the glycerol molecules released during the breakdown of triacylglycerides being used for gluconeogenesis by the liver [27,28]

Previous attempts to profile metabolites associated with cachexia have yielded varying results and differences in important metabolites produced in each study. Metabolomics research in cachexia began in 2008 in the C26 mouse model [5]. This was the first study to demonstrate a distinct metabolomics-based profile associated with the onset of muscle wasting and identify increased levels of very low and low density lipoprotein associated with aberrant glycosylation of β-Dystroglycan (β-DG, a marker of muscle wasting in this mouse model) [29]. Human metabolomics studies in cancer cachexia have investigated urine and plasma from weight-losing patients. These studies also found large numbers of glycerophospholipids and metabolites associated with amino acid metabolism and were able to identify occult sarcopenia in patients with cancer [6,30]. Recent studies have attempted to separate pre-cachectic, cachectic and weight stable cancer patients and healthy controls using serum metabolomics, and enabled identification of fifteen highly discriminative metabolites [8]. The present study was able to discriminate using weight-losing from weight-stable patients using only six metabolites. The only study to find a markedly different metabolic pattern to the present study analysed patients using three analytical platforms, namely gas chromatography mass spectrometry, capillary electrophoresis mass spectrometry, and LC/MS. The authors found a significant reduction in amino acids and glycerophospholipids associated with cachexia, a difference that has not previously been described in this condition, plus a high increase in cortisol levels [9]. All bar one of these studies [6] used 5% weight loss as a cut off to stratify patients, in a similar fashion to the present study. Future studies should investigate the relationship between metabolomics and dynamic assessments of tissue loss over time, e.g., using serial CT body composition analysis.

Blood-based metabolomics is a promising method for cachexia research. However, as seen previously, results are often difficult to replicate due to the heterogeneity of the populations and study sizes. All the patients included in the present study were suitable for potentially curable surgical resection of their cancer and therefore had localized/non-metastatic disease and similar measures of muscle volume, indicating that the identified metabolites might be early markers of fat wasting preceding muscle loss. This hypothesis is supported by the fact that two of the patients in the WS group were “cachectic” according to the consensus definition, by virtue of having ≥2%WL and low SMI on CT, and yet, they did not group with the identified panel of six metabolites. This finding may be explained by a lack of dynamic wasting, and that CT-identified sarcopenia may be the pre-cancer patient norm, rather than a consequence of disease.

One obvious limitation of this study is the small number of samples used, involving differences in sex ratios and cancer types between the groups. This was an exploratory study involving patients without refractory cachexia and was not designed to identify sex- or tumour-specific differences. This study was limited to patients with UGI cancers and therefore, may not be applicable to all cancer types. Previous serum metabolomic studies in pancreatic and oesophageal cancer have confirmed that dysregulation of lipid metabolism is a key component of these conditions, although the exact tissue site of these processes is unknown [31,32]. In the present study, differences in tumour type between patient groups are unlikely to explain the identified six-metabolite signature, due to the close correlation with patient weight loss. We were able to demonstrate distinct metabolic profiles consistent with the consensus cachexia definition of ≥5% weight loss. These findings support our previous muscle transcriptomic studies, and they give further biological validation to the 2011 Fearon consensus definition of cancer cachexia [1] as a valid patient inclusion criterion in clinical trials. Furthermore, these results provide a 6-metabolite profile for further investigation as a marker of cachexia in longitudinal studies, with the opportunity to explore early diagnosis and response to therapeutic intervention.

## 4. Materials and Methods

### 4.1. Participants

Patients were all over 18 years of age and were recruited from the regional upper gastrointestinal (GI) multi-disciplinary team meeting. Written informed consent was obtained from all subjects and ethical approval received from Lothian Research Ethics Committee (UK, ethic code: 06/S1103/75) Participating patients had a diagnosis of upper GI cancer (oesophageal, gastric, pancreatic or duodenal) and were undergoing surgery with the intent of curative resection of the primary tumour. All patients had normal kidney function. Clinical details, degree of weight loss from self-reported pre-illness stable weight, and body mass index (BMI) were recorded.

### 4.2. CT Body Composition Analysis

Skeletal muscle cross-sectional area (CSA) was measured from routine staging CT scans performed prior to any surgical intervention. A transverse CT image from the third lumbar vertebrae (L3) was assessed for each scan date and tissue volumes estimated using semi-automated software. Cross-sectional area for muscle, subcutaneous and visceral adipose tissue was normalized for stature (cm^2^/m^2^) to calculate the skeletal muscle, subcutaneous and visceral adipose indices (SMI, SATI and VATI respectively).

### 4.3. Sample Collection and Storage

Fasting venous blood samples were taken at induction of anaesthesia approximately four to six weeks after the cessation of any neoadjuvant chemotherapy. Samples were allowed to clot at room temperature. Serum was separated by centrifugation at 1300 RPM for 12 min at a temperature of 20 degrees Celsius. C-reactive protein (CRP) was measured in Clinical Chemistry, Royal Infirmary, Edinburgh (fully accredited by Clinical Pathology Accreditation Ltd., UK) using standard automated methods. A CRP ≥5 mg/L was considered consistent with the presence of systemic inflammation. Samples were stored locally at −80 °C until transported to the metabolomic facility (Strathclyde Institute of Pharmacy and Biomedical sciences) in cool bags at −30 °C.

### 4.4. Chemicals and Solvents

HPLC grade Acetonitrile (ACN) was purchased from Fisher Scientific (Loughborough, UK) and HPLC grade water was produced by a Direct-Q3 UltrapureWater System (Millipore, Watford, UK). AnalaR-grade formic acid (98%) was obtained from BDH-Merck (Poole, UK). Authentic stock standard metabolites (Sigma-Aldrich, Poole, UK) were prepared as previously described and diluted four times with ACN before LC-MS analysis. Ammonium acetate was purchased from Sigma-Aldrich (Poole, UK).

### 4.5. Sample Preparation

Exactly 200 µL of the sample was mixed with 800 µL ACN containing 10 µg/mL of 2 ^13^C glycine (Sigma-Aldrich) as an internal standard to ensure retention time stability, then centrifuged for 10 min before transferring into a vial with an insert. In order to prepare the QC sample, 0.05 mL of plasma was taken from each of the samples and mixed in order to make a pooled sample. The pooled sample was prepared by pipetting 50 µL from each of the 18 samples and then mixing them together before diluting 0.2 mL of the pooled sample with 0.8 mL ACN containing 5 µg/mL of 2 ^13^C glycine internal standard and centrifuging. Additionally, the prepared mixtures of authentic standard metabolites containing 10 µg/mL of 2 ^13^C glycine as internal standard were run.

### 4.6. LC-MS Conditions

Liquid chromatographic separation was carried out on an Accela HPLC system interfaced to an Exactive Orbitrap mass spectrometer (Thermo Fisher Scientific, Bremen, Germany) using both a ZIC-pHILIC column (150 × 4.6 mm, 5 µm, HiChrom, Reading, UK). The column was eluted with a mobile phase consisting of 20 mM ammonium carbonate in HPLC-grade water (solvent A) and ACN (solvent B), at a flow rate of 0.3 mL/min. The elution gradient was an A:B ratio of 20:80 at 0 min, 80:20 at 30 min, 92:8 at 35 min and finally, 20:80 at 45 min. The nitrogen sheath and auxiliary gas flow rates were maintained at 50 and 17 arbitrary units. The electrospray ionisation (ESI) interface was operated in both positive and negative modes. The spray voltage was 4.5 kV for positive mode and 4.0 kV for negative mode, while the ion transfer capillary temperature was 275 °C. Full scan data were obtained in the mass-to-charge ratio (m/z) range of 75 to 1200 for both ionisation modes on the LC-MS system fully calibrated according to the manufacturer’s guidelines. The resulting data were acquired using the XCalibur 2.1.0 software package (Thermo Fisher Scientific).

### 4.7. Data Extraction and Analysis

Patients were grouped into two and analysed based on percentage weight loss (> or <5%). Data extraction for each of the samples was carried out by MZMatch software. The extracted ions, with their corresponding m/z values and retention times, were pasted into an Excel macro of the most common metabolites prepared in-house to facilitate identification. The lists of the metabolites obtained from these searches were then carefully evaluated manually by considering the quality of their peaks and their retention time match with the standard metabolite mixtures run in the same sequence. All metabolites were within 3 ppm of their exact masses. The list of metabolites was refined by removing all metabolites with RSD > 20% in the pooled samples, leaving a list of 318 metabolites. Statistical analyses were performed using both univariate with Microsoft Excel and multivariate approaches using SIMCA-P software version 14.1 (Umetrics, Umea, Sweden). The 318 metabolites were modelled to give a PCA plot and then supervised analysis based on OPLS was carried out by refining the list of metabolites by eliminating the less important variables to give a model with the lowest possible number of variables.

## 5. Conclusions

These results show that metabolomic profiles in plasma from cancer patients are different between patients with ≥ or <5% weight loss. Most of the metabolites identified within these profiles fell within the lipid pathways. Differences highlighted in the breakdown of lipids provide an understanding of the mechanisms involved in the pathogenesis of cachexia. A six-metabolite signature correlated strongly with degree of patient weight loss. A better understanding of the importance of adipose wasting and the potential sharing of datasets between studies may identify novel biomarker strategies and therapeutic approaches for cancer cachexia.

## Figures and Tables

**Figure 1 cancers-11-01594-f001:**
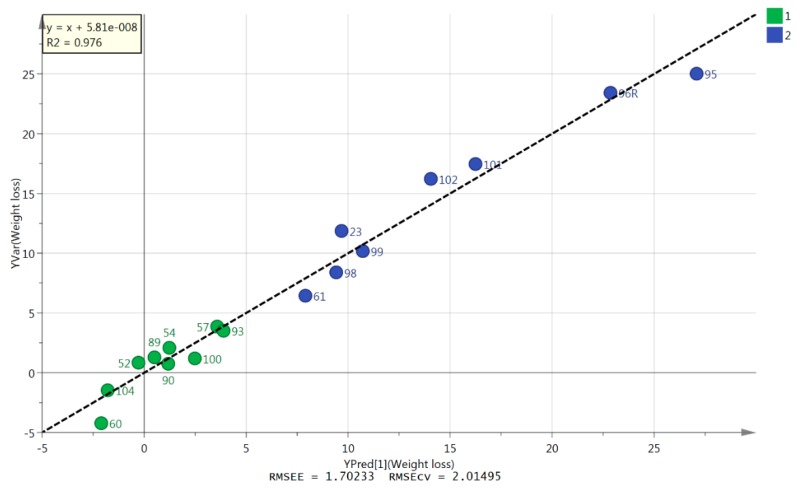
Orthogonal partial least squares (OPLS) model. OPLS model showing close correlation for eight ≥5% WL samples and nine WS samples between predicted and actual weight change (CVANOVA = 0.0014) based on six variables listed in Table 2. Green circles (group 1) = WS, blue circles (group 2) = ≥5% WL.

**Figure 2 cancers-11-01594-f002:**
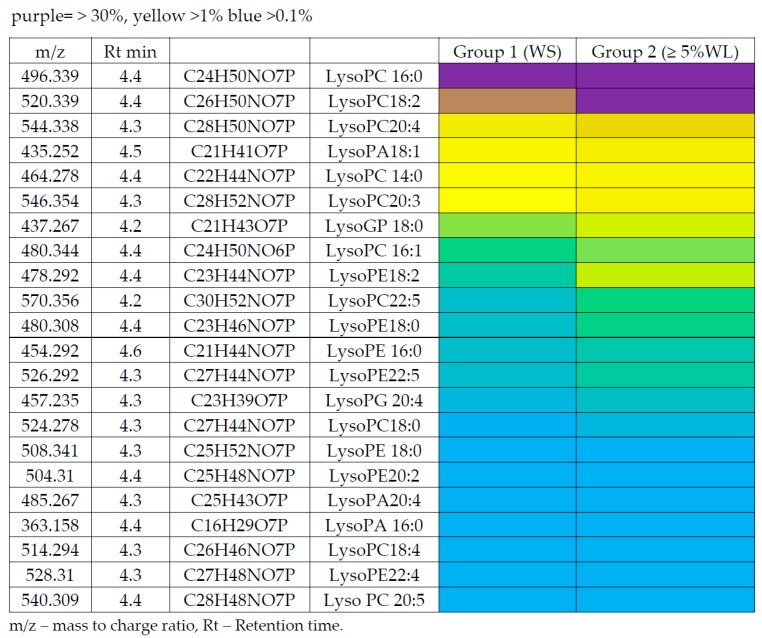
Heat map showing relative levels of lysolipids.

**Table 1 cancers-11-01594-t001:** Patient details.

Demographics	Group 1 Weight Stable (*n* = 9)	Group 2 ≥5% Weight Loss (*n* = 9)	*p* Values
Male: Female	8:1	5:4	N/A
Age (years)	61 (4.65)	66 (10.53)	0.167
% Weight loss	2.13 (1.35)	14.39 (6.56)	0.001 *
SMI	47.17 (6.26)	45.82 (7.72)	0.536
SATI	46.25 (20.38)	58.43 (33.86)	0.379
VATI	57.57 (55.28)	42.10 (33.48)	0.506
BMI (kg^2^/m^2^)	24.93 (4.42)	26.29 (4.64)	0.534
CRP (mg/L)	17.88 (27.06)	32.56 (50.44)	0.453
Cancer type	Pancreatic – 1	Pancreatic – 6	N/A
	Oesophageal – 6	Oesophageal – 2	
	Gastric - 2	Duodenal - 1	
Disease stage	1 (*n* = 1)	1 (*n* = 1)	
	2 (*n* = 0)	2 (*n* = 5)	
	3 (*n* = 6)	3 (*n* = 1)	
	4 (*n* = 1)	4 (*n* = 2)	
	Unknown (*n* = 1)	-	
Pre-operative chemotherapy	4	2	N/A

All data are mean (standard deviation). SMI—Skeletal muscle index, SATI—subcutaneous adipose tissue index, VATI—Visceral adipose tissue index, BMI—Body mass index, CRP = C Reactive protein. * Denotes statistically significant result.

**Table 2 cancers-11-01594-t002:** The six metabolites used to produce the OPLS model shown in Figure 1

m/z	Rt Min.	Metabolite	VIP Value
520.339	4.4	Lyso-PC 18:2	1.82
116.071	13.0	L-Proline	1.43
255.233	4.3	Hexadecanoic acid	0.54
281.249	3.8	Octadecenoic acid	0.42
166.086	10.0	Phenylalanine	0.36
480.344	4.4	Lyso-PC 16:1	0.20

M/Z—Mass to charge ratio, Rt—Retention time, VIP—Variable importance in projection.

**Table 3 cancers-11-01594-t003:** Significant metabolites that differ between ≥ 5% WL and WS groups (*n* = 9 and 9 respectively).

Polarity	m/z	Rt(min)	Metabolite	*p* Value	Ratio WL/WS
Amino acids					
P	116.071	13.0	L-Proline	0.015	1.36
P	166.086	10.0	L-Phenylalanine	0.619	0.87
Fatty acids					
N	214.048	4.3	sn-glycero-3-Phosphoethanolamine	0.006	1.78
N	255.233	4.3	Hexadecanoic acid	0.049	1.21
N	277.217	3.9	Octadecatrienoic acid	0.010	1.60
N	279.233	4.2	Linoleate	0.002	1.36
N	281.249	3.8	Octadecenoic acid	0.023	1.22
N	293.249	4.2	Nonadecadienoic acid	0.019	1.24
N	303.233	4.1	Eicosatetraenoic acid	0.022	1.37
N	305.249	4.2	Eicosatrienoic acid	0.054	1.50
N	327.233	4.2	Docosahexaenoic acid	0.025	0.81
N	329.249	4.1	Docosapentaenoic acid	0.009	1.46
N	331.264	3.9	Docosatetraenoic acid	0.014	1.68
P/N	380.255	5.1	Sphingenine phosphate	0.033	1.28
Lipids					
N	214.048	4.3	Glycerophosphoethanolamine	0.006	1.78
N	381.205	4.6	LPA 14:0	0.010	1.73
N	393.241	4.4	LPA 16:0 ether	0.048	1.37
N	433.236	4.4	LPA 18:2	0.001	1.67
N	435.252	4.5	LPA18:1	0.006	1.40
N	437.267	4.2	LPA 18:0	0.028	1.23
P/N	454.292	4.6	LPE 16:0	0.040	1.44
N	457.235	4.3	LPA 20:4	0.001	1.62
N	464.278	4.4	LPC 14:1	0.007	1.36
P	468.308	4.6	LPC 14:0	0.026	1.58
P/N	476.278	4.4	LPE 18:2	0.013	1.98
P/N	478.292	4.4	LPE 18:1	0.013	2.02
P/N	480.308	4.4	LPE 18:0	0.012	2.07
P/N	480.344	4.4	LPC16:1	0.089	1.32
N	485.267	4.3	LPA 22:4	0.007	1.96
P/N	496.339	4.4	LPC 16:0	0.040	1.34
N	498.262	4.3	LPE 20:5	0.053	2.13
P/N	500.278	4.4	LPE 20:4	0.002	2.36
N	504.31	4.4	LPE 20:2	0.002	1.65
N	514.294	4.3	LPC 18:4	0.005	2.28
P/N	520.339	4.4	LPC 18:2	0.001	1.75
P/N	524.278	4.3	LPE 22:6	0.032	1.45
N	526.294	4.3	LPE 22:5	0.004	2.51
N	528.31	4.3	LPE22:4	0.004	2.01
P/N	544.338	4.3	LPC 20:4	0.014	1.81
P/N	546.354	4.3	LPC 20:3	0.026	1.91
P	570.356	4.2	LPC 22:5	0.009	1.89
P	731.605	4.2	SMd18:0/18:1	0.052	1.29

M/Z = Mass to charge ratio, WS= Weight stable, WL = Weight losing, PE = phosphatidyl ethanolamine, PC = phosphatidyl choline, PA = phosphatidic acid, L = lyso, P = Detection in positive ion mode, N = Detection in negative ion mode.

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
