# Peer review of "Plasma Metabolomics Identifies Lipid and Amino Acid Markers of Weight Loss in Patients with Upper Gastrointestinal Cancer"

_cancers, 2019, doi:10.3390/cancers11101594_

Round 1

Reviewer 1 Report

All comments were addressed

Reviewer 2 Report

The revised manuscript is well corrected.

Reviewer 3 Report

looked at paper and responses — the paper is acceptable as is in its revised form

This manuscript is a resubmission of an earlier submission. The following is a list of the peer review reports and author responses from that submission.

Round 1

Reviewer 1 Report

I like this research and greatly appreciate the work going into this.

We need a better understanding of cancer cachexia.   I have really only few question / critique.   1. When 6 factors are identified with current presence of weight loss >5% and they are not elevated in patients without such weight loss, how can these markers then ever be used as "use in early diagnosis”, presumably predicting future weight loss?  For this, one would need biomarkers that are raised,, BEFORE weight loss has developed and that redict future presence of weight loss >5%. Is any such data available?   2. Also – and this is an obvious limitation of the study – the study is not only small, but it is limited to patients with upper GI cancer of not too severe kind (i.e. with a good enough status to permit surgery with currative ambition).  This limitation (i.e. that this is research done in patients with upper GI cancer only) should be added in the conclusion part of the paper (as it is appropriately done in the title).   3. What about the impact of kidney function on plasma levels of these metabolites? maybe body wasting affects kidneys function and then %weight loss strongly relates to levels of these markers —> can this be excluded as a problem? is eGFR available for patients?  

Reviewer 2 Report

This study by Miller et al. nicely shows the difference in metabolomics profiles of plasma between patients with >= or < 5% WT loss, as the authors stated in the "Conclusions". However, several points remain to inconclusive from the results of the study, some of which may be fundamental.

1. The most important limitation is the small number of samples, as the authors stated in the last paragraph in the "Results" section. The reviewer is afraid that the most important imbalance may be the type of cancer, where most of the patients with WT loss had pancreatic cancer. Readers may consder that the difference in plasma metabolomics did not detect WT loss but pancreatic cancer. Several metabolites shown to be associated with WT loss were those regarding fatty acid or lipid metabolism. However, it may be unclear if abnormality in metabolites regarding fatty acid or lipid metabolism reflect pathophysiology in cachexia or abnormalities caused by pancreatic cancer. One possible way to discriminate this issue may be sub-analysis of association between WT change and metabolimics focused on 7 patients with pancreatic cancer. Another way may be to validate the result in another cohort.

2. Although the authors emphasize that definition of cancer cachexia by Fearon et al., data in the present study do not include the length of period in which WT loss was documented. The definition by Fearon et al. indecates WT loss in " 6 months".

Reviewer 3 Report

The manuscript by Miller et al. aims at investigating the metabolomics profiling of serum samples from patients affected with  gastrointestinal cancers to the extent of identifying markers of cachexia. The goal of the study is definitely important, as markers for such debilitating complication of cancer are notoriously lacking; hence identifying a marker would certainly have a significant impact on the field. However, at the present stage several major weaknesses in study design and statistical analysis reduce the excitement for this manuscript and must to be addressed before being accepted for publication.

Study design – major concerns:

Page 2, line 64. The study appears to be prospective, although the authors do not indicate why they recruited only 18 patients, nor do the authors indicate if patients were consecutive (cross sectional) or purposefully picked (case control). A major weakness is represented by the fact that the samples used in this manuscript derive from patients which, despite being cachectic based on the currently accepted definition of cachexia, are not showing changes in SKI. Keeping in mind evidence generated by Vickie Baracos and Carla Prado, SKI is a more reliable indicator of cachexia and well correlates with survival in patients with cancer, compared to BMI and body weight. For this reason, this reviewer believes the analysis might be inconclusive, as no changes in SKI are reported among the patients enrolled in the study. This is also further corroborated by the absence of complete separation between the two groups, as shown in the PCA plot (Figure 1). The study might be underpowered. Indeed, only 18 subjects were recruited, including patients with different sex and diagnoses. Moreover, it is not mentioned whether the patients were undergoing chemotherapy treatments, previously described to cause cachectic phenotypes and metabolic alterations, at time of blood sampling. It is interesting to notice that despite a reduction in BW, all other parameters, including SMI, SATI, VATI, BMI and CRP, are unchanged. All these parameters would seem once again to support the idea that no real cachexia is evident in such patients. Moreover, the authors fail to describe the amount of time it took patients to lose weight (by definition, it usually has to be over six months). Only the blood is investigated for the presence of biomarkers that could be indicative of cachexia. However, cachexia is known to be a multi-systemic and multi-organ comorbidity of cancer, hence investigating other tissues may further corroborate these findings. On this regards, it would also be interesting to know about the treatment status of these patients. A recent metabolomic profiling conducted in animals bearing cancers  and exposed to chemotherapy (Pin et a. 2019 JCSM) suggest differential effects and further corroborate the idea that chemotherapeutics may directly contribute to the occurrence of a phenotype consistent with cachexia. The authors fail to describe clinical features of the patients that could contribute to the weight loss. For example, what if all the patients in the weight loss group have very large metabolically active tumors or other comorbidities such as heart failure or COPD? Moreover, they may have tumors that inhibit digestion in some way (bowel obstructive, etc) which would lead to malnutrition and not cachexia necessarily. We assume they do not have metastases since they are having elective surgery, but they fail to provide a clinical picture that would avoid confounding their results. (table 1) Accordingly, it would be interesting to read about the nutritional status of these subjects. The statistical analysis may be flawed. Indeed, the authors need to at least perform a univariate analysis to look at whether these variables influence their results. I find this particularly so since the weight stable group mainly has esophageal cancer and the cachexia group mainly has pancreatic cancer. (table 1) They also do not test for homogeneity within the individual groups. Moreover, I find it curious that the authors state that there was a patient in the weight loss group whose data matched the weight stable group and therefore they just removed the results (Page 6, line 129). What is the scientific rationale for excluding this data?

Statistical analysis - major concerns:

The PCA plot does not show any separation, hence I am not sure it makes any sense to even show it in the main paper. Would be interesting to see the PCA as a supplementary figure, with the inclusion of the QC samples so that we can evaluate the stability of the plaform. There is no description of how the QC samples were prepared. Were they pooled from all samples? The OPLS-DA plots of the whole set and the reduced set should be shown and each should contain the necessary validation data e.g. R2Y and Q2Y. They should also include how many components were used to model the data.  Table 2 reports 6 metabolites used to generate the OPLS model. How were the 6 metabolites selected for the reduced model? From Table 2, only the top two metabolites have VIP values > 1.  This is a typical metric for significant explanation of the Y variable. Interestingly, Phenylalanine is included in the reduced model despite the low VIP value and, as indicated in Table 3, the p-value is non-significant (0.619). Please explain. Lyso-PC 16:1 is also included in the reduced model, but is not listed in Table 3 as one of the significantly altered metabolites. Please explain. The information content of Figure 3 is not clear. A more routinely used heatmap constructed of the z-scores of metabolite values of each group would be more informative. Line 131. This sentence indicates that the test would be 95% accurate.  It’s not clear how this number was derived.  A more appropriate evaluation of test accuracy would be an AUC analysis which is readily computed in Simca.